# Sweet, Salty, and Umami Taste Sensitivity and the Hedonic Perception of Taste Sensations in Adolescent Females with Anorexia Nervosa

**DOI:** 10.3390/nu14051042

**Published:** 2022-02-28

**Authors:** Magdalena Hartman-Petrycka, Ewa Klimacka-Nawrot, Katarzyna Ziora, Wanda Suchecka, Piotr Gorczyca, Katarzyna Rojewska, Barbara Błońska-Fajfrowska

**Affiliations:** 1Department of Basic Biomedical Science, Faculty of Pharmaceutical Sciences in Sosnowiec, Medical University of Silesia, 40-055 Katowice, Poland; ewakn@sum.edu.pl (E.K.-N.); wandas@sum.edu.pl (W.S.); bbf@sum.edu.pl (B.B.-F.); 2Department of Paediatrics, Faculty of Medical Sciences in Zabrze, Medical University of Silesia, 40-055 Katowice, Poland; kziora@sum.edu.pl; 3Department of Psychiatry, Faculty of Medical Sciences in Zabrze, Medical University of Silesia, 40-055 Katowice, Poland; pgorczyca@sum.edu.pl; 4Pediatric Endocrinology, Public Clinical Hospital No.1, 41-800 Zabrze, Poland; krojewska@szpital.zabrze.pl

**Keywords:** anorexia nervosa, recognition thresholds, taste intensity, hedonic response, sweet, salty, umami

## Abstract

Objective: The aim of this study was to perform analysis of sensitivity to sweet, salty, and umami tastes based on three measurement methods and of the hedonic perception of taste sensations in adolescent females with anorexia nervosa (AN). The aim of the research was to confirm the results of other authors in terms of the perception of sweet and salty taste in patients with AN, and then develop knowledge about the perception of umami taste, which is still insufficiently studied. Method: A total of 110 females with an age ranging from 13 to 19 years, including 50 newly diagnosed patients with a restrictive subtype of AN and 60 healthy controls participated in gustatory research involving analyses of taste perception (recognition thresholds, ability to identify the taste correctly, taste intensity, and hedonic response) applying the sip and spit method. Results: Females with AN showed reduced sensitivity to salty taste and increased sensitivity to umami taste and, more often than healthy controls, wrongly classified the taste of solutions with a low sucrose concentration. Patients with AN assessed the sodium chloride and monosodium glutamate tastes less negatively than did control participants, and they did not show differences in their hedonic assessment of sucrose. Conclusions: The taste sensitivity alterations in females with AN demonstrated in this paper do not entail decreased hedonic assessment of taste experiences. Based on our results, we cannot consider the observed variation in taste sensitivity in patients with AN to be a factor that increases their negative attitude toward food consumption.

## 1. Introduction

Inappropriate eating behaviors are one of the main symptoms of anorexia nervosa (AN). Prolonged malnutrition and endocrine disorders resulting from starvation may lead to dysgeusia. However, there are limited published studies devoted to taste sensitivity and the hedonic perception of taste in patients with AN. The results of research on this issue are ambiguous. Several researchers have demonstrated decreased taste sensitivity in people with AN [1,2,3,4]. According to others, in patients affected by AN, sensitivity to sweet taste remains unchanged [5,6,7,8,9,10]. Taste sensitivity has a significant influence on food intake control, and its evaluation in patients with AN may provide valuable information about disease development and treatment progress. Nozoe et al. [4] observed that a rapid improvement in taste responsiveness during therapy is accompanied by a positive therapeutic effect, i.e., energy consumption at the level of 1600 kcal/day. The authors believe that early improvement in taste responsiveness suggests decreased resistance to an applied therapeutic program. The pleasure found in taste sensations has a considerable influence on the motivation to eat. 

The methodology of the study taste perception in patients with AN presented in the literature is very diverse [11]. Despite the differences in the test group selection, there are major differences in the methodology of taste tests, such as the tastes used, the method of applying the tastes, and the taste modality tested. The most frequently studied taste was the sweet taste, in 12 publications [4,5,6,7,8,9,10,12,13,14,15,16]; mainly, taste recognition, taste intensity, and taste detection were assessed. The hedonic aspect of the sweet taste was described less frequently in the literature, and its results were contradictory. Some authors indicated that hedonic perception of sweet taste in patients with AN does not differ compared to those without AN, whereas others [12,17] notice a decrease in the hedonic ratings of sweet test stimuli. In this paper, attempts were made to confirm the results presented in the literature regarding sweet taste perception in patients with AN using four methods of its assessment (recognition thresholds, ability to correctly identify the taste, taste intensity, and hedonic taste perception).

The salty taste perception in patients with AN is the subject of less research [2,4,8,9,13,14,15,18], and the most complete analysis, including the hedonic assessment of sodium chloride solutions, is presented in only one publication [16]. The analysis of the salty taste perception in patients with AN presented in this paper will help to complete the knowledge in this area.

The perception of umami taste in patients with AN was examined only in one publication [13], in which no significant differences were found in the recognition threshold, intensity of perception or hedonic assessment of this taste. Considering the small number of tested persons (15 patients with AN and 15 healthy subjects) and a small number of test samples (6 concentrations), the results presented in Goldzak-Kunik et al.’s publication [13] require confirmation.

The aim of this study was to perform an assessment of taste sensitivity to sweet, salty, and umami tastes based on recognition thresholds, ability to identify the taste correctly, taste intensity, and on the hedonic perception of taste sensations in adolescent females with AN. The results of the study may be useful in determining the role of taste perception in the development of eating disorders such as AN.

## 2. Material and Methods

### 2.1. Participants

The participants were 110 females ranging in age from 13 to 19 years, including 50 newly diagnosed patients who met the criteria for a restrictive subtype of AN (AN group), according to the Diagnostic and Statistical Manual of Mental Disorders, 5th edition (DSM-5) [19] classification, and 60 healthy volunteers constituting the control group (C group). The body mass index (BMI) of the AN group ranged from 11.1 to 18.3 kg/m^2^ (mean 15.0 ± 1.7 kg/m^2^), and the mean age was 15.3 ± 1.6 years. BMI values in the C group ranged from 18.6 to 24.9 kg/m^2^ (mean 20.8 ± 1.6 kg/m^2^), and the mean age was 15.9 ± 1.4 years. The anthropometric characteristics of the study participants are presented in Table 1.

The volunteers were selected to participate in gustatory testing after pediatric, psychological, and psychiatric consultations. The control participants were in overall good physical and mental health and did not take any medicines or hormonal contraceptives. Patients with AN were excluded if they had current or lifetime medical disorders with the potential to affect food consumption and weight (e.g., diabetes mellitus or thyroid disorders, gastrointestinal or neurological disorders, salivary dysfunctions, etc.) or if they were on any medicines at the time of testing. All patients with AN suffered from secondary amenorrhea. They were hospitalized at the Department of Pediatrics and Children’s Endocrinology at the University Hospital in Zabrze, which specializes in the treatment of eating disorders. Realimentation was introduced on the first day of hospitalization, and the gustatory tests were performed within the following three days. 

The study was conducted in accordance with the Helsinki Declaration, and each participant and her parent or legal guardian provided written consent after having been informed of the aim, protocol, and methodology of the study. The research project was approved by the Bioethics Committee of the Medical University of Silesia (decision ID: KNW-6501-62/08). All participants gave their written consent to participate in the study. The legal guardians of underage girls also gave their written consent to participate.

### 2.2. Procedure 

The test was carried out in a sensory analysis laboratory meeting the requirements listed in ISO 8589 [20]. Our gustatory research consisted of analyses of taste perception of sweet, salty, and umami tastes. For taste perception, we used the following aqueous solutions (Table 1): sucrose for sweet, sodium chloride for salty, and monosodium glutamate for umami. All samples were labeled with randomized three-digit codes. The participants had no knowledge of the types of substances used or the coding system. The patients tasted the samples in the order shown in Table 2, i.e., at increasing concentrations, to minimize the effect of taste adaptation.

They were asked to rinse their mouth with distilled water before tasting the sets consisting of 10 or 3 samples, and not to rinse their mouth between tasting the samples of a given set. A standard sip and spit procedure was employed. The concentration of solutions complied with the standards of ISO 3972 [21]. Based on the results of the studies conducted at our department for several years, we have decided that the concentration range should be increased by adding two higher concentrations than those recommended (numbers 9 and 10 in Table 2). The patients held a solution in their mouths, completed the ratings, and expectorated the solution. An entire 15-mL sample was taken into the mouth and spit out after 5 s. The results were recorded by the participants on specifically designed assessment sheets. Taste sensations in response to each sample from a series of 10 concentrations were evaluated by the patients with respect to quality (no sensation, sweet, salty, bitter, sour, or umami). Based on these results, a taste recognition threshold (the lowest concentration resulting in the correct recognition of a taste in relation to its quality) was assigned. Additionally, based on the same results, the ability to correctly identify the taste was assessed. The percentage of samples in each group in which the taste was not recognized, in which the taste was recognized correctly, and in which it was indicated that the substance had a taste, but the wrong name was used to define it, was calculated. For example, when using sodium chloride solutions, test subjects incorrectly indicated that they sensed a sweet, sour, umami, or bitter taste. For each series of dilutions, the percentage of erroneous recognition was calculated with the exact description of which taste was incorrectly indicated; additionally, the total incorrect taste recognition was calculated as the sum of all incorrect taste indications. The lack of taste and correct taste recognition was not included in total incorrect taste recognition. 

A three-sample series of suprathreshold concentrations was used to examine both the intensity and the hedonic perception of each taste sensations. The females recorded the intensity of taste perceptions on a 10 cm linear analog scale, with the starting point marked “0” and the end point marked “10” (maximum taste perception). The degree of pleasure derived from a taste sensation was indicated on a linear analog scale, with the extreme points described as maximally unpleasant (−5.0) and maximally pleasant (+5.0) and the middle point as neutral (0). Exemplary linear analog scales for the first sample of sodium chloride are presented in Figure 1. Similar scales were used for the remaining suprathreshold concentrations of sodium chloride, sucrose and monosodium glutamate, with appropriate modification of the tastes above the lines.

The results were obtained by measuring the distance from the zero point on the scale to a subject’s mark. The tests were performed in the morning, and the participants were in a fasting state.

### 2.3. Statistical Analysis

A statistical analysis of the data was performed using Microsoft Excel 2007 and Statistica 9.0 software and involved the following tests: the Shapiro–Wilk test, used to evaluate the normality distribution of the data, the Mann–Whitney U test, used to evaluate the anthropometric parameters, the taste recognition threshold, the intensity of experience, and hedonic perception; the Chi-square test, used to compare correct and incorrect taste recognition between the AN group and controls; and the Spearman’s rank correlation coefficient, used to assess the relationships between BMI and taste response (intensity and hedonic perception). A *p* value <0.05 was considered statistically significant. 

## 3. Results

The taste recognition thresholds for sucrose and sodium chloride did not differ between the AN group and controls, but for monosodium glutamate, the threshold was lower in the AN group (Median: 8, 1st Quartile: 5, 3rd Quartile: 11) compared with the C group (Median: 10, 1st Quartile: 7, 3rd Quartile: 11) (*p* < 0.05) (Figure 2). 

Both in the group of patients with AN and in the C group, there were cases of incorrect assessment of the tastes of the presented solutions, labeled in Table 2 with numbers from 1 to 10 in the range from 0.16 g/L to 4.07 g/L for sodium chloride, in the range from 0.34 g/L to 33.33 g/L for sucrose, and in the range from 0.08 g/L to 2.04 g/L for monosodium glutamate. The percentage of erroneous qualitative assessments of the taste of sodium chloride solutions did not differ significantly between the AN and the C groups. However, we found intergroup differences in the taste assessment of sucrose and monosodium glutamate solutions (Table 3).

A larger percentage of incorrect qualitative assessments of sucrose solutions (*p* < 0.05) was noted in the AN group (37.2%) compared with the C group (30.3%), whereas the taste of monosodium glutamate solutions was less frequently incorrectly assessed in the AN group (45.8%) than in the C group (54.5%) (*p* < 0.01). The percentage of correct assessments of the taste of monosodium glutamate solutions was higher in the AN group than in the C group (34.0% vs. 19.5%; *p* < 0.001), and the patients with AN less often had no sensation of the taste of monosodium glutamate solutions than did the C group (20.2% vs. 26.0%; *p* < 0.05). The females with AN did not sense the taste of sodium chloride in samples more often than the C group (29.8% vs. 23.2%; *p* < 0.05) and showed a tendency to less frequently correctly assess the taste of sucrose solutions (38.4% vs. 43.7%; *p* = 0.077). 

The taste assessments of sodium chloride solutions of concentrations 0.18%, 0.36%, and 0.90% were different between the two groups (Figure 3A,B).

The median (Me) and the first quartile (Q1) and the third quartile (Q3) of taste intensity of 0.18% sodium chloride concentrations in AN and C groups were, respectively, Me: 1.6 and 1.7, Q1: 0.4 and 1.1, and Q3: 2.3 and 3.6. (*p* < 0.05) (Figure 3A). The taste intensity of 0.36% sodium chloride concentrations in AN and C groups were, respectively, Me: 6.0 and 6.3, Q1: 2.3 and 5.1, and Q3: 7.4 and 8.5. (*p* < 0.05). The taste intensity of 0.90% sodium chloride concentrations in AN and C groups were, respectively, Me: 9.1 and 9.8, Q1: 8.0 and 8.5, and Q3: 9.8 and 10.0 (*p* < 0.05). The taste of the highest concentration of sodium chloridethe solution in patients with AN was less unpleasant than in C group (Figure 3B). The hedonic responses to 0.90% sodium chloride in the AN and C groups were Me: −2.5 and −4.1, Q1: −4.6 and −5.0, and Q3: 0.5 and −1.2, respectively (*p* < 0.05).

The analysis of the intensity and hedonic perception of taste sensations evoked by sucrose in the AN and C groups showed no significant differences (Figure 4A,B).

Similarly, no significant differences in the evaluation of the taste intensity of monosodium glutamate solutions of concentrations: 0.1%, 0.3%, and 1.0% were discovered (Figure 5A). However, significant differences in the hedonic perception of other taste sensations were found. The taste of the 1% monosodium glutamate solution was described as less unpleasant by the AN group compared with the C group. The hedonic responses to 1.0% monosodium glutamate in AN and C groups were Me: 0.0 i −3.4, Q1: −3.0 i −4.9, and Q3: 2.0 i 0.0 (*p* < 0.001), respectively. A similar trend in the hedonic responses to 0.3% monosodium glutamate in AN and C groups were observed (Me: −0.2 i −1.4, Q1: −2.5 i −3.1, and Q3: 0.0 i 0.0) (*p* = 0.064) (Figure 5B).

The data analysis did not show any correlation between BMI and taste response (recognition threshold, intensity, and hedonic perception) in the group of patients with AN or in the C group (Table 4).

## 4. Discussion

Taste sensation assessment has received little attention in the scientific literature, although it presumably plays an important role in eating disorders. Given that taste perception and related disorders can be assessed with respect to qualitative, quantitative, and hedonic aspects, the present study aimed to investigate the taste recognition thresholds for the provided examples of taste substances, ability to identify the taste correctly and the intensity and hedonic perception of taste sensations in patients with AN. We used solutions of sodium chloride, sucrose, and monosodium glutamate for this purpose. 

Although the taste recognition threshold for sodium chloride did not demonstrate any abnormal taste sensitivity in the AN group, the intensity with which patients with AN experienced the taste of solutions with suprathreshold concentrations of sodium chloride was decreased. This change may influence the perception of food consumed. It was noticed, however, that the ability of the patients with AN to assess the quality of salty taste did not differ from that of the controls, which suggests that the AN group’s qualitative perception of salty taste in food was not distorted.

Our results demonstrate that taste sensitivity determined by the recognition threshold for and the intensity of sensation of the sucrose taste did not differ between the AN group and the controls. However, it was noted that the females with AN more often incorrectly assessed the taste of sucrose than the females without eating disorders did. The taste may cause patients with AN to experience a sensation of bitter taste and may change their perception of what they eat. It needs to be stressed that the cases, observed both in the AN and in the C groups, of incorrect qualitative perception of sweet taste mainly applied to low sucrose concentrations that were above the sensation threshold defined as the weakest stimulus that an organism can sense and below the recognition threshold defined as the lowest concentration, resulting in the correct recognition of a taste in relation to its quality. 

It is not easy to test umami taste recognition because the name “umami” is still not familiar to Polish people, and the use of this expression was difficult for the subject, despite our prior explanations preparing the participants for identification of this taste. The participants with AN showed a better ability to recognize umami taste than the healthy participants did, as evidenced by a taste recognition threshold for monosodium glutamate that was lower than that in the C group, the lower percentage of incorrect qualitative assessments, and the lower percentage of cases of no sensation of monosodium glutamate taste. The better ability of the patients with AN to correctly recognize umami taste confirmed their higher sensitivity to this taste quality. However, this ability may also result from the fact that the participants were able to better use the information about umami taste that they had been given by us before the tests and that they were better at using the word “umami”. Females with AN are characterized by high ambitions and striving for perfection [22]. It is difficult, however, to explain the smaller percentage of cases with no taste sensation for monosodium glutamate in the AN group by referring to the patients’ character traits because experiencing or not experiencing a taste is unambiguously associated with the sense of taste. To summarize, patients with AN were, compared to controls, characterized by a lower umami taste recognition threshold, better ability to identify the umami taste correctly, and indicate the taste of the 1% monosodium glutamate solution as less unpleasant, while they did not differ from the control group in intensity of perception of the umami taste sensation. In the studies conducted by Goldziak-Kunik et al. [13], no statistically significant differences in the intensity of umami taste perception in girls with AN compared to healthy individuals have been shown. Slight differences in the hedonic assessment of taste in our results and Goldziejak-Kunik et al. [13] may come from methodological differences, such as the different size and characteristics of the study groups, and the use of different monosodium glutamate concentrations. The current literature does not describe the umami taste recognition threshold in girls with AN and it is not possible to compare the obtained results with the results of other authors. 

While analyzing the literature examining the taste perception in people with AN, depending on the method used to test taste, hypogeusia, or the absence of taste disturbances dominates. Unfortunately, the research methodology is not homogeneous, and it is still difficult to draw final and binding conclusions [11].

There are a few published analyses of taste sensitivity in people with AN that have focused on the separate evaluation of particular tastes. Assessment of sensitivity to sweet taste in patients with AN appears to be especially interesting, explaining why most attention has been paid to this matter in the literature. Despite data indicating that AN is accompanied by sweet taste hypogeusia (higher recognition threshold, filter paper disc method) [4] (reduced detection, filter paper disc method) [14] other research [5,6,7,8,9,10,12,13] has concluded that this condition does not involve sweet taste perception, which is the same in patients with AN and in people without eating disorders. Therefore, our results corroborate these researchers’ findings. 

Salty taste hypogeusia in the literature has been demonstrated by Nozoe et al. (higher recognition threshold, filter paper disc method) [4], Nakai et al. (reduced detection, filter paper disc method) [14], and Jirik-Babb et al. (lower intensity rating, no difference in taste recognition, measuring magnitude estimation at different concentrations in distilled water) [15]. The studies presented in this manuscript showed a decrease in the intensity of salty taste perception in girls with AN, but the recognition threshold for this taste did not differ from the results obtained among the healthy control. In the publications assessing the perception of salty taste in the AN group, no disturbance of salty taste perception appears [2,8,9,13,18].

Apart from the qualitative and quantitative aspects of taste sensitivity in AN, hedonic assessment of taste experiences seems most important. It has been proven that the results of hedonic perception studies are dependent on the method of taste substance application, i.e., whether the testing solution is swallowed or spat out [21]. It is believed that these differences result from the fear of gaining weight experienced by people with AN [23]. The current study, applying the sip and spit method, showed that the adolescents with AN did not differ from the participants without eating disorders in their hedonic responses to the taste of sucrose. Our findings have confirmed studies by Simon et al. [7] and Klein et al. [24]. However, Sunday et al. [12] described a decrease in the hedonic response to the sweet taste of sucrose in patients with AN, although according to the studies by Drewnowski et al. [5] and Simon et al. [7], fat aversion, and not a fear of carbohydrates, is the key factor in food selection by patients with AN. 

Given that the feeling of pleasure is a motivating factor shaping behavior, we can consider a measure of pleasure derived from taste perception as a measure of appetite. In people without eating disorders, appetite is an important factor influencing the quality and quantity of food intake and thus eating habits. In the current paper, we studied whether eating disorders in females with AN are connected with an abnormal hedonic response to a taste. Our results showed that the hedonic response to the taste of solutions with suprathreshold concentrations of sodium chloride or monosodium glutamate in the AN group was higher than that in the C group, whereas the hedonic response to the taste of sucrose solutions was the same in both groups. Therefore, the avoidance of food consumption by females with AN is not associated with disorders of the hedonic response to a taste sensation.

The results of studies using functional magnetic resonance imaging (fMRI) indicate that brain reward circuits are more responsive to food stimuli in AN and that the brains of individuals who have recovered from AN are characterized by aberrant neural responses to pleasant and aversive taste stimuli [24,25,26,27] examined women who had recovered from AN and showed that the disturbances of gustatory processing in the central nervous system do not necessarily coexist with aberrations in subjective taste sensations.

In this study, the relationships between BMI and sweet, salty, and umami tastes (recognition thresholds, intensity, and hedonic assessment of taste) were assessed in girls with AN and in girls from the control group. No significant correlations were found, which is consistent with the results of the study conducted by Nozoe et al. [4]. However, in Aschenbrenner et al. [1], a positive relationship between the taste sensitivity measured with the taste strip test kit and BMI was found. It is worth emphasizing, however, that in the studies by Aschenbrenner et al. [1] patients with AN were much older (mean ± standard deviation; 24.5 ± 4.0 years), while the newly diagnosed patients with AN aged 15.3 ± 1.6 years participated in this study. Perhaps the deterioration in taste, along with a decrease in BMI, occurs with a longer duration of the disease.

### Limitations

The methodology of the perception of taste sensations studies functioning in the world of science is very diverse. Different tastes are used to induce the taste stimulus, different methods of taste application, and a different range of taste concentrations. It is also important which aspect of the perception of taste sensations is assessed. Within the scope of sensitivity of taste perception alone, there are three aspects of the research: sensation/detection threshold, recognition threshold, and intensity of taste perception. Moreover, hedonic responses to taste sensations can also be specified. Tasting substances are most often applied in the form of filter paper discs, water solutions, and reference meal [11].

In this study, we tried to investigate the perception of taste sensations of a given taste, hence the assessment of the recognition threshold, the intensity of taste perception, and its hedonic responses. Based on the pilot studies, the range of concentrations was extended to assess recognition thresholds, and the intensity of perception and hedonic responses were carried out on additional solutions with suprathreshold concentrations of taste substances. The use of as many as 13 solutions to test each taste makes the presented research results more accurate and includes observations that are not so widely analyzed by other authors. The research methodology could still be improved. It would be beneficial to ask for taste qualities before the hedonic assessment and to eliminate the name of the taste from the scale. Such a procedure would additionally identify people with difficulties in recognizing the category of tastes in supra-threshold concentrations and its impact on the hedonic assessment.

The most frequently assessed in patients with AN taste qualities are sweet, salty, bitter, and sour, also known as basic tastes [11]. In this study, the perception of sweet and salty tastes was assessed, and umami was also included, which is also a basic taste, but is the least studied in the context of patients with AN. The assessment of the three tastes in a wide range means that, due to physiological reasons, sensory adaptation, and a decrease in concentration, it was not possible to introduce bitter and sour tastes into the research, which is a significant limitation of this work.

In this study, the sip and spit method was used, this method is more accurate than the filter paper disc method because the taste substance dissolved in water comes into contact with all receptors present in the oral cavity. At the same time, it is necessary to remember that the perception of aqueous solutions of tastes does not fully reflect into the real perception of meals, because nutritional behaviors also depend on texture, smell, appearance, and even portion size.

The results were obtained after examining a large number of girls with AN (*n* = 50) with the participation of the control group (*n* = 60). The influence of BMI on taste perception was analyzed, which is an advantage of this study. At the same time, only the restrictive subtype of AN was included with the omission of other types of disorder, which limits the conclusions.

## 5. Conclusions

In this paper, we demonstrated several differences in taste perception between females with restrictive subtype AN and females without eating disorders. Changed taste sensitivity in females with AN most certainly influences their perception of what they eat. However, the less negative hedonic assessment of the suprathreshold concentrations of sodium chloride and monosodium glutamate and the lack of differences in the hedonic assessment of sucrose solutions indicate that the observed differences in the perception of salty, sweet, and umami taste are not the key factors that contributed to negative attitude toward eating. 

## Figures and Tables

**Figure 1 nutrients-14-01042-f001:**
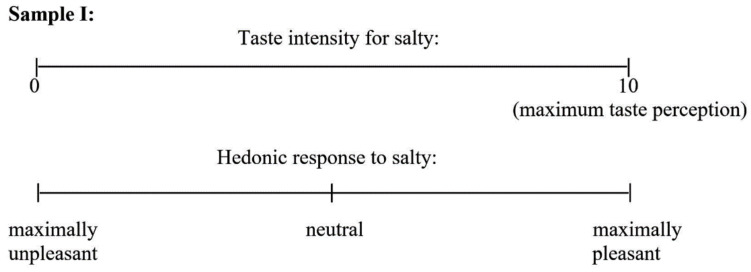
Part of the answer sheet used for taste intensity and hedonic responses to 0.18% sodium chloride.

**Figure 2 nutrients-14-01042-f002:**
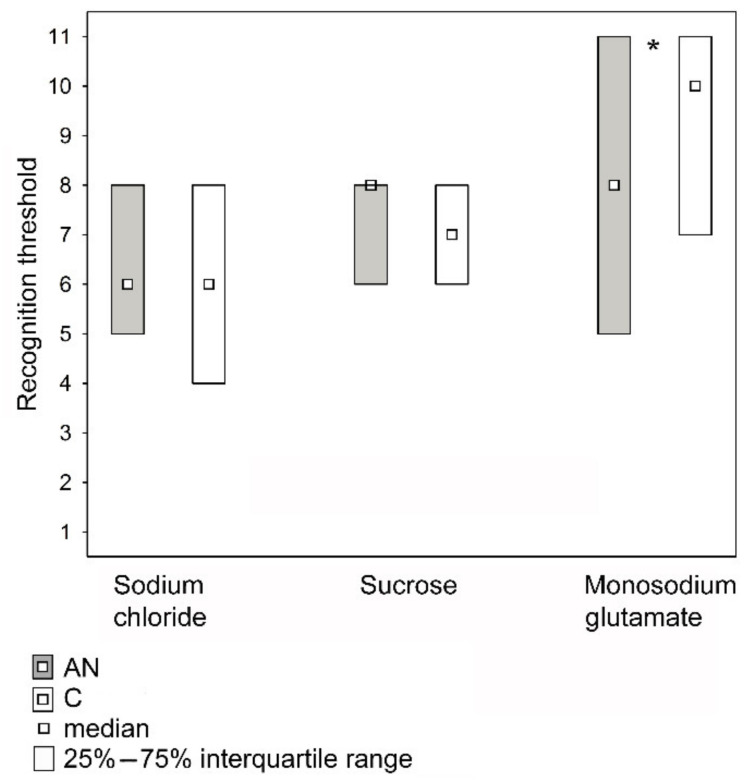
Recognition thresholds for the sodium chloride, sucrose, and monosodium glutamate tastes in patients with anorexia nervosa (AN; *n* = 50) and in healthy individuals (C; *n* = 60). Median: Sodium chloride AN = 6 (0.98 g/L), C = 6 (0.98 g/L), Sucrose AN = 8 (12.0 g/L); C = 7 (7.20 g/L), Monosodium glutamate AN = 8 (1.0 g/L), C = 10 (2.04 g/L), * *p* < 0.05 (the Mann–Whitney U test).

**Figure 3 nutrients-14-01042-f003:**
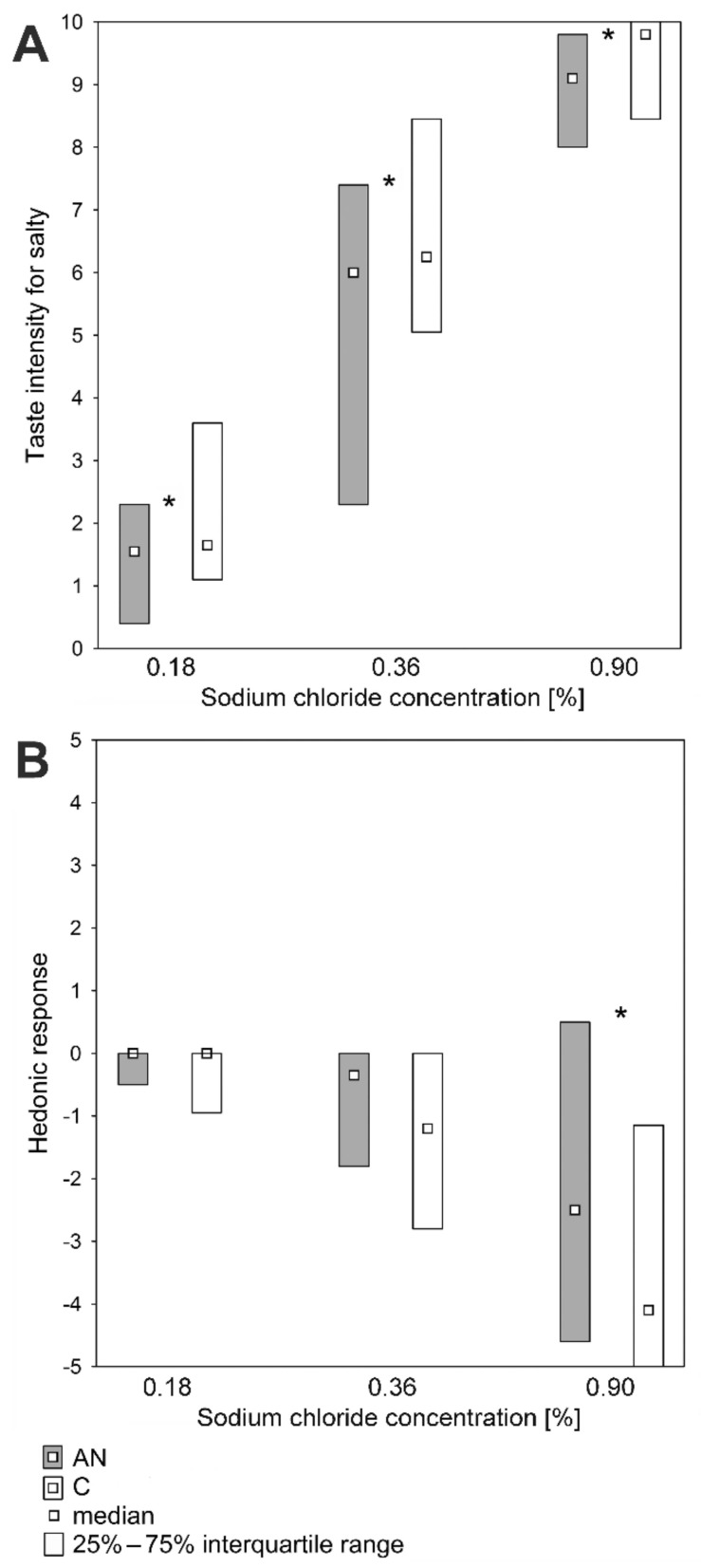
Salty taste perception in patients with anorexia nervosa (AN; *n* = 50) and in healthy individuals (C; *n* = 60). (**A**) Suprathreshold intensity for salty taste. (**B**) Hedonic responses to sodium chloride solutions. * *p* < 0.05 (the Mann–Whitney U test).

**Figure 4 nutrients-14-01042-f004:**
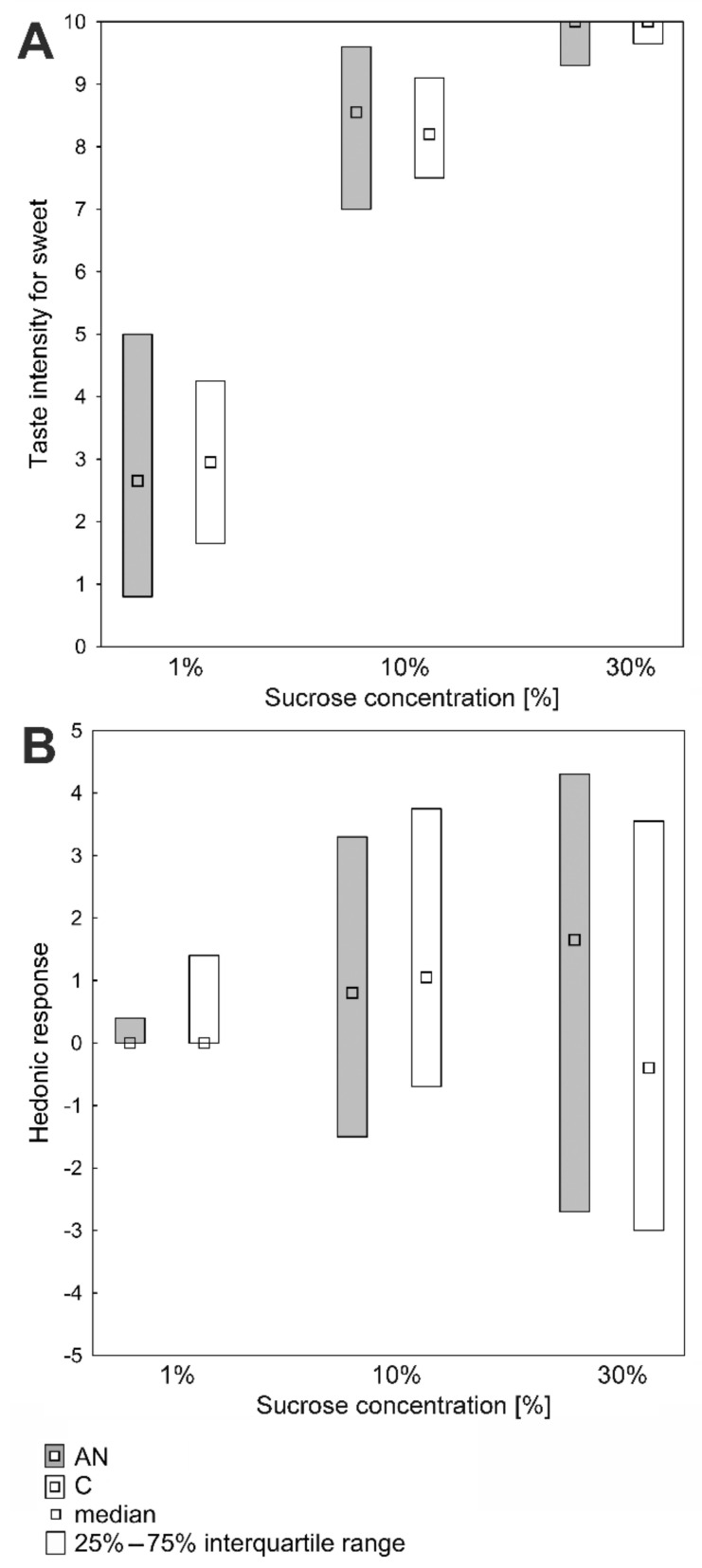
Sweet taste perception in patients with anorexia nervosa (AN; *n* = 50) and in healthy individuals (C; *n* = 60). (**A**) Suprathreshold intensity for sweet taste. (**B**) Hedonic responses to sucrose solutions (the Mann–Whitney U test).

**Figure 5 nutrients-14-01042-f005:**
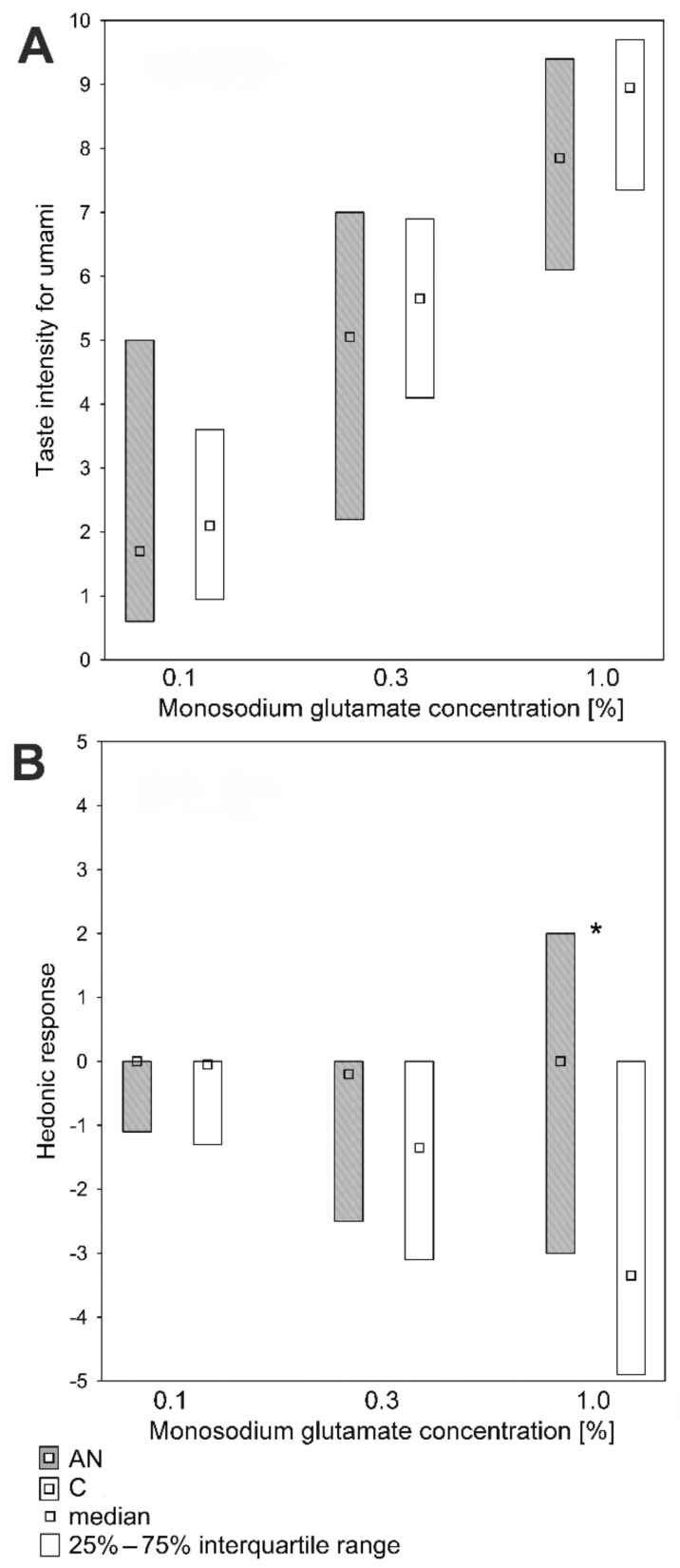
Umami taste perception in patients with anorexia nervosa (AN; *n* = 50) and in healthy individuals (C; *n* = 60). (**A**) Suprathreshold intensity for umami taste. (**B**) Hedonic responses to monosodium glutamate solutions. * *p* < 0.001 (the Mann–Whitney U test).

**Table 1 nutrients-14-01042-t001:** Anthropometric characteristics of the study participants.

	Mean		Standard Deviation		Median		Minimum		Maximum	
	AN	C	AN	C	AN	C	AN	C	AN	C
Body weight [kg] ***	39.32	56.41	5.96	5.67	38.75	56.30	26.80	45.60	52.40	73.50
Height [m] *	1.62	1.65	0.07	0.06	1.62	1.65	1.46	1.50	1.77	1.77
BMI [kg/m^2^] ***	15.01	20.77	1.72	1.63	15.03	20.34	11.12	18.65	18.25	24.89

AN: Patients with AN (*n* = 50), C: Healthy controls (*n* = 60), *** *p* < 0.001, * *p* < 0.05 (the Mann–Whitney U test).

**Table 2 nutrients-14-01042-t002:** Concentrations of taste substances in the solutions used for gustatory testing.

Order of Solutions Presented	Concentration			Type of Examination
	Sodium Chloride	Sucrose	Monosodium Glutamate	
	(g/L)	(g/L)	(g/L)	Taste sensitivity (recognition threshold) and ability to correctly identify the taste
1	0.16	0.34	0.08	
2	0.24	0.55	0.12	
3	0.34	0.94	0.17	
4	0.48	1.56	0.24	
5	0.69	2.59	0.34	
6	0.98	4.32	0.49	
7	1.40	7.20	0.70	
8	2.00	12.00	1.00	
9	2.85	20.00	1.43	
10	4.07	33.33	2.04	
	(%)	(%)	(%)	Taste perception of suprathreshold concentrations (taste intensity and hedonic responses)
I	0.18	1	0.1	
II	0.36	10	0.3	
III	0.90	30	1.0	

**Table 3 nutrients-14-01042-t003:** Taste sensations in response to all samples from a series of 10 concentrations of sodium chloride, sucrose, and monosodium glutamate.

Taste Sensations (Ability to Identify the Taste Correctly)	Sodium Chloride		Sucrose		Monosodium Glutamate	
	AN	C	AN	C	AN	C
Salty	42.6%	47.3%	4.4%	4.5%	13.0%	12.3%
Sweet	5.8%	8.8%	38.4%	43.7%	4.0%	5.2%
Umami	6.8%	6.0%	6.0%	1.2%	34.0% ***	19.5%
Sour	5.2%	5.8%	8.0%	10.3%	11.2%	14.2%
Bitter	9.8%	8.8%	18.8%	14.3%	17.6%	22.8%
No taste sensation	29.8%	23.2%	24.4%	26.0%	20.2%	26.0%
Total incorrect taste recognition	27.6%	29.4%	37.2% *	30.3%	45.8% **	54.5%

The gray color means incorrect taste recognition. * *p* < 0.05, ** *p* < 0.01, and *** *p* < 0.001 (the Chi-square test). The results were obtained from patients with AN (AN; *n* = 50) and healthy controls (C; *n* = 60).

**Table 4 nutrients-14-01042-t004:** Correlation between BMI and recognition thresholds, taste intensity, and hedonic responses in the group of patients with AN (*n* = 50) and in the C group (*n* = 60) (the Spearman’s rank correlation).

		AN		C	
		R	*p*	R	*p*
Recognition threshold for sodium chloride		−0.17	0.24	0.18	0.18
Taste intensity for sodium chloride	0.18%	−0.11	0.46	0.00	1.00
	0.36%	0.10	0.51	−0.15	0.27
	0.90%	0.19	0.18	0.01	0.91
Hedonic responses to sodium chloride	0.18%	−0.15	0.29	0.02	0.89
	0.36%	−0.10	0.49	−0.05	0.69
	0.90%	−0.02	0.89	−0.12	0.35
Recognition threshold for sucrose		0.09	0.53	0.17	0.19
Taste intensity for sucrose	1%	0.21	0.14	0.03	0.84
	10%	0.14	0.32	−0.04	0.76
	30%	0.03	0.84	0.03	0.85
Hedonic responses to sucrose	1%	−0.10	0.48	−0.24	0.07
	10%	0.02	0.87	0.08	0.53
	30%	−0.02	0.91	−0.10	0.45
Recognition threshold for monosodium glutamate		0.20	0.17	0.21	0.10
Taste intensity for monosodium glutamate	0.1%	0.06	0.68	−0.02	0.91
	0.3%	−0.09	0.55	−0.02	0.91
	1.0%	0.02	0.89	0.01	0.96
Hedonic responses to monosodium glutamate	0.1%	−0.12	0.40	0.08	0.57
	0.3%	−0.12	0.39	0.09	0.51
	1.0%	−0.15	0.31	0.05	0.68

## Data Availability

The datasets used and/or analyzed during the current study are available from the corresponding author on reasonable request.

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
