# Peer review of "Sweet, Salty, and Umami Taste Sensitivity and the Hedonic Perception of Taste Sensations in Adolescent Females with Anorexia Nervosa"

_nutrients, 2022, doi:10.3390/nu14051042_

Round 1

Reviewer 1 Report

The etiology of aberrant food intake in anorexia nervosa (AN) is unclear; thus this study addresses an important issue. 

Strengths of the study include: (a) a large cohort of subjects with AN; (b) a healthy control group; (c) the blinding of subjects to the components of the test solutions; and (d) a systematic approach to studying taste perception, admittedly a difficult subject.

Issues that must be addressed:

1. It appears that taste sensitivity was assessed using solutions that contain NaCL, sucrose and monoNa glutamate.  Did the authors test solutions containing single components alone?

2. The authors need to clarify the issue of "incorrect assessments"; the meaning of this (how exactly was this assessed?) is unclear.

3. Figure 4 as shown is identical to Figure 3 - the monoNa glutamate data are not presented

4. the authors must clarify the difference between "sensation threshold" and "recognition threshold"

5. lines 247-8: you mean that your findings confirm those of Simon et al and Klein et al

6. line 262: the phrase "may confirm the psychological origin of the disease" should be eliminated

Author Response

Dear Reviewer,

We would like to thank you for positive review. Your comments were very useful and helped to improve our manuscript.

  1. It appears that taste sensitivity was assessed using solutions that contain NaCL, sucrose and monoNa glutamate.  Did the authors test solutions containing single components alone?

Response: Yes, the single components: sodium chloride, sucrose or monosodium glutamate were tested.

  1. The authors need to clarify the issue of "incorrect assessments"; the meaning of this (how exactly was this assessed?) is unclear.

Response: Description was added. Lines: 145-154

  1. Figure 4 as shown is identical to Figure 3 - the monoNa glutamate data are not presented

 Response: The correct figure was added. Line: 248-249

  1. The authors must clarify the difference between "sensation threshold" and "recognition threshold"

Response: Definitions were added. Line: 282-285

  1. Lines 247-8: you mean that your findings confirm those of Simon et al and Klein et al.

Response: Sentence was corrected. Line: 343-344

  1. Line 262: the phrase "may confirm the psychological origin of the disease" should be eliminated

Response: Sentence was deleted. Line: 359

Reviewer 2 Report

The study by Hart-Petrycka et al investigated the taste thresholds and hedonic rating of anorectic vs healthy control females towards the taste qualities sweet, umami and salty. The study design seems appropriate, as does the number of subjects used for the study. The necessary ethical aspects are clarified in the manuscript.

In general, the novelty and originality of the study has to be rated as rather low, as several studies already looked into taste responses in anorectic persons. Since no clear research gap or a hypothesis is mentioned in the manuscript, the added value of the study is a descriptive study on the topic of AN and taste that supports previous findings. In addition, in the version I received, the data showing one of the most important finding of the study is lacking (two times the same figure with sucrose included instead).

Therefore, several aspects need to be considered before publication in Nutrients can be considered:

  • Define the research gap that can be answered with your study and clearly state this, including a clear hypothesis, in the introduction as well as in the abstract
  • The term “complex analysis” should be defined. I would expect some type of multivariate analysis here or or a repeated-measures ANCOVA to define covariates? Three out of five classical taste qualities in a two-group comparison seems a bit weak for the term complex
  • The participant characteristics should be listed in a table including a statistical analysis in which aspects the groups where different
  • Were the sensory tests performed in a sensory laboratory under controlled surrounding? Include information on the test setting (temp control, noise level, humidity, light…)
  • Was a power analysis performed? Why 50 vs 60 subjects? Why only include female subjects?
  • Was the hormone status of the control persons (phase of menstrual cycle) included?
  • Why was not bitter or sour taste included in the analysis?
  • Why were only non-parametric statistical tests applied? Were there any tests for normality and equal variety carried out before?
  • The figure legends should include the information of the statistical analysis and the number of subjects included in the analysis.
  • 128: I do have problems to understand the sentence, or find the referenced data in table 1. Please rephrase/ clarify
  • Figure 3 and Figure 4 are the same. Figure 4 is missing.
  • 225. please correct sentence (I assume Kinnaird et al. should be excluded here?)
  • 234.235 sentence includes spelling mistakes (at al), comma etc.
  • Include in the result section the most important result, e.g. state the mean / median threshold with standard deviation for both groups so that the reader gets a better sense of the data
  • In the introduction and the discussion section, more details about the cited studies are required à subjects, conditions etc is lacking, but needed to judge methodological differences
  • 244 a reference for this statement is needed
  • 247 “Studies by Simon et al. [7] and Klein et al. [22] have confirmed our 247 findings.” à please correct: your study confirmed the results of the previous studies, not the other way round. Your study came later-
  • The limitations of the study are not discussed à please in include all limitations and strengths of the study in the discussion sections
  • The statistical section mentioned Spearman correlations, but I didn’t find a reference in the text?
  • It would be of interest to investigate a relation of the taste data to the severity of AN, BMI, in between the ratings. I think the manuscript would highly benefit from including those data, as this can increase the originality a lot. See also my second comment, maybe a repeated measures ANCOVA can be considered (should be possible with n=50).
  • Title for the manuscript needs to be specified, this is too general for the outcome of the study
  • “we need to emphasize that the taste sensitivity alterations demonstrated in this paper seem to be the result of disease progression because the alterations do not entail a decrease in the hedonic response to taste sensations.” – How can this statement be made in light of the shown data?

Author Response

Dear Reviewer,

We would like to thank you for positive review. Your comments were very useful and helped to improve our manuscript.

  1. The study by Hart-Petrycka et al investigated the taste thresholds and hedonic rating of anorectic vs healthy control females towards the taste qualities sweet, umami and salty. The study design seems appropriate, as does the number of subjects used for the study. The necessary ethical aspects are clarified in the manuscript.

Response: Description was added. Lines: 117-118

  1. In general, the novelty and originality of the study has to be rated as rather low, as several studies already looked into taste responses in anorectic persons. Since no clear research gap or a hypothesis is mentioned in the manuscript, the added value of the study is a descriptive study on the topic of AN and taste that supports previous findings. In addition, in the version I received, the data showing one of the most important finding of the study is lacking (two times the same figure with sucrose included instead).

Therefore, several aspects need to be considered before publication in Nutrients can be considered:

Define the research gap that can be answered with your study and clearly state this, including a clear hypothesis, in the introduction as well as in the abstract

Response: Description was added. Lines: 19-21; 57-86

  1. The term “complex analysis” should be defined. I would expect some type of multivariate analysis here or or a repeated-measures ANCOVA to define covariates? Three out of five classical taste qualities in a two-group comparison seems a bit weak for the term complex

Response: The term “complex analysis” refers to a large number of measurment methods of taste perception, not statistical methods. Description was converted. Lines: 17-18; 81

  1. The participant characteristics should be listed in a table including a statistical analysis in which aspects the groups where different

Response: Table 1 was added. Lines: 99-101

  1. Were the sensory tests performed in a sensory laboratory under controlled surrounding? Include information on the test setting (temp control, noise level, humidity, light…)

Response: Description was added. Lines: 120-121

  1. Was a power analysis performed? Why 50 vs 60 subjects? Why only include female subjects?

Response: Due to the lack of a normal distribution of the empirical data, no power analysis was performed. During the study, efforts were made to recruit as many volunteers as possible, and the final number was determined by practical considerations. Anorexia nervosa is a disease that primarily affects young girls. Although boys were also sent to the hospital ward, they were many times smaller than girls, on average there was 1 boy per 25 girls. In order not to hinder the interpretation of the results due to the selection of the study group, it was decided not to include boys in the study.

  1. Was the hormone status of the control persons (phase of menstrual cycle) included?

Response: No

  1. Why was not bitter or sour taste included in the analysis?

Response: Description was added. Lines: 387-401

  1. Why were only non-parametric statistical tests applied? Were there any tests for normality and equal variety carried out before?

Response: Description was added. Lines: 164-165

  1. The figure legends should include the information of the statistical analysis and the number of subjects included in the analysis.

Response: Information to all figure legends were added.

  1. 128: I do have problems to understand the sentence, or find the referenced data in table 1. Please rephrase/ clarify

Response: Description was added. Lines: 184-186

  1. Figure 3 and Figure 4 are the same. Figure 4 is missing.

Response: The correct figure was added. Line: 248-249

  1. please correct sentence (I assume Kinnaird et al. should be excluded here?)

Response: Sentence was corrected. Line: 317

  1. 235 sentence includes spelling mistakes (at al), comma etc.

Response: Sentence was corrected. Line: 328

  1. Include in the result section the most important result, e.g. state the mean / median threshold with standard deviation for both groups so that the reader gets a better sense of the data

Response: Descriptions were added. Lines: 174-175; 184-186; 212-221; 237-242

  1. In the introduction and the discussion section, more details about the cited studies are required à subjects, conditions etc is lacking, but needed to judge methodological differences

Response: Descriptions were added. Lines: 57-80; 308-312; 322-323; 328-331; 367-376

  1. 244 a reference for this statement is needed

Response: Reference was added. Line 340

  1. 247 “Studies by Simon et al. [7] and Klein et al. [22] have confirmed our 247 findings.” à please correct: your study confirmed the results of the previous studies, not the other way round. Your study came later-

Response: Sentence was corrected. Line: 343-344

  1. The limitations of the study are not discussed à please in include all limitations and strengths of the study in the discussion sections

Response: Limitation was added. Lines: 377-412

  1. It would be of interest to investigate a relation of the taste data to the severity of AN, BMI, in between the ratings. I think the manuscript would highly benefit from including those data, as this can increase the originality a lot. See also my second comment, maybe a repeated measures ANCOVA can be considered (should be possible with n=50).

Response: Due to the lack of a normal distribution, we use non-parametric tests. The ANCOVA test should not be used in this situation. Tables with the results of the correlation between BMI and individual variables of the perception of taste sensations have been added. Lines: 255-258; 367-376

  1. Title for the manuscript needs to be specified, this is too general for the outcome of the study

Response: Title was changed. Lines: 2-5

  1. “we need to emphasize that the taste sensitivity alterations demonstrated in this paper seem to be the result of disease progression because the alterations do not entail a decrease in the hedonic response to taste sensations.” – How can this statement be made in light of the shown data?

Response: Conclusion was changed. Lines: 412-427

Round 2

Reviewer 1 Report

Lines 80-85

The aim of this study was to perform a complex an assessment of taste sensitivity to sweet, salty and umami tastes based on recognition thresholds, ability to identify the taste correctly, taste intensity and on the hedonic perception of taste sensations in adolescent females with AN. The aim of the research was to confirm the results of other authors in terms of the perception of sweet and salty taste in patients with AN, and the develop knowledge about the perception of umami taste, which is still not studied enough.

Suggest that you eliminate the second sentence

Lines 366-368

In this study, the results of the sweet, salty and umami taste (recognition thresholds, intensity and hedonic assessment of taste) were correlated with the BMI value in girls with AN and in girls from the control group. No significant relations were found.

Suggest you change to”

In this study, the relationships between BMI and sweet, salty and umami taste (recognition thresholds, 366 intensity and hedonic assessment of taste) were assessed in girls with AN and in girls from the control group. No significant correlations were found.

Lines 370-71

a dated relationship between the taste sensitivity measured with the taste strip test kit and BMI was found.

What is “a dated relationship”?

Author Response

Thank you very much for all your comments.

Lines 80-85

The aim of this study was to perform a complex an assessment of taste sensitivity to sweet, salty and umami tastes based on recognition thresholds, ability to identify the taste correctly, taste intensity and on the hedonic perception of taste sensations in adolescent females with AN. The aim of the research was to confirm the results of other authors in terms of the perception of sweet and salty taste in patients with AN, and the develop knowledge about the perception of umami taste, which is still not studied enough.

Suggest that you eliminate the second sentence

Response: The sentence has been eliminated

Lines 366-368

In this study, the results of the sweet, salty and umami taste (recognition thresholds, intensity and hedonic assessment of taste) were correlated with the BMI value in girls with AN and in girls from the control group. No significant relations were found.

Suggest you change to”

In this study, the relationships between BMI and sweet, salty and umami taste (recognition thresholds, 366 intensity and hedonic assessment of taste) were assessed in girls with AN and in girls from the control group. No significant correlations were found.

Response: The sentence has been changed

Lines 370-371

a dated relationship between the taste sensitivity measured with the taste strip test kit and BMI was found.

What is “a dated relationship”?

Response: the term was corrected to „positive relationship”

Reviewer 2 Report

The authors improved the manuscript according to the reviewers' comments which significantly improved the overall quality.

Author Response

Thank you very much for the reviews.